# Mitochondrial morphodynamics alteration induced by influenza virus infection as a new antiviral strategy

Irene Pila-Castellanos[1,2�], Diana Molino[2�], Joe McKellar[3], Laetitia Lines[1], Juliane Da Graca[2], Marine Tauziet[3], Laurent Chanteloup[1], Ivan Mikaelian[4], Laurène Meyniel-Schicklin[1], Patrice Codogno[2], Jacky Vonderscher[1], Cédric Delevoye[5,6], Olivier Moncorgé[4], Eric Meldrum[1], Caroline Goujon[4], Etienne Morel[2]*, Benoit de Chassey[1]*

**1** ENYO-Pharma, Lyon, France, **2** Institut Necker-Enfants Malades (INEM), INSERM U1151-CNRS UMR 8253, Université de Paris, Paris, France, **3** Institut de Recherche en Infectiologie de Montpellier (IRIM), UMR 9004—CNRS, Université de Montpellier, Montpellier, France, **4** Université de Lyon, Université Claude Bernard Lyon 1, INSERM 1052, CNRS 5286, Centre Léon Bérard, Centre de recherche en cancérologie de Lyon, Lyon, France, **5** Institut Curie, PSL Research University, CNRS, UMR144, Structure and Membrane Compartments Paris, France, **6** Institut Curie, PSL Research University, CNRS, UMR144, Cell and Tissue Imaging Facility (PICT-IBiSA), Paris, France

ய These authors contributed equally to this work.
* etienne.morel@inserm.fr (EM); bdc@enyopharma.com (BC)

**Data Availability Statement:** All relevant data are within the manuscript and its Supporting Information files.

## Abstract

Influenza virus infections are major public health threats due to their high rates of morbidity and mortality. Upon influenza virus entry, host cells experience modifications of endomembranes, including those used for virus trafficking and replication. Here we report that influenza virus infection modifies mitochondrial morphodynamics by promoting mitochondria elongation and altering endoplasmic reticulum-mitochondria tethering in host cells. Expression of the viral RNA recapitulates these modifications inside cells. Virus induced mitochondria hyper-elongation was promoted by fission associated protein DRP1 relocalization to the cytosol, enhancing a pro-fusion status. We show that altering mitochondrial hyper-fusion with Mito-C, a novel pro-fission compound, not only restores mitochondrial morphodynamics and endoplasmic reticulum-mitochondria contact sites but also dramatically reduces influenza replication. Finally, we demonstrate that the observed Mito-C antiviral property is directly connected with the innate immunity signaling RIG-I complex at mitochondria. Our data highlight the importance of a functional interchange between mitochondrial morphodynamics and innate immunity machineries in the context of influenza viral infection.

## Author summary

Influenza virus infections cause significant diseases and socio-economic burden. The current therapeutic arsenal is restricted to drugs that essentially target two proteins of the virus. In this study, we investigated endomembrane modifications inside cells following influenza virus infection. We find remarkable elongation of mitochondria associated with

**Funding:** This work was supported by institutional funding from INSERM, CNRS, University Paris-Descartes and the French National Research Agency through the "Investments for the Future" program (France-BioImaging, ANR-10-INSB-04, C.D.)., The PICT IBiSA is supported by the Cell(n) Scale Labex (ANR-10-LBX-0038, C.D.) part of the IDEX PSL (ANR-10-IDEX-0001-02 PSL, C.D.). The funders had no role in study design, data collection and analysis, decision to publish, or preparation of the manuscript.

**Competing interests:** The authors have declared that no competing interests exist.

a reduction in the number of contact sites between mitochondria and endoplasmic reticulum, platforms known to be critical for innate immunity regulation. We demonstrated that the sole expression of a fragment of the viral genome is sufficient to provoke these modifications and we identified how the main drivers of the mitochondria fusion/fission machinery behave to favor such an elongated state. We introduce potential application of Mito-C, a new drug that inhibits influenza virus replication by counteracting these membrane modifications. We finally demonstrated that the functional result of this action is a booster of the innate immune response of the cell. Thus, Mito-C has a broad spectrum potential to fight other RNA viruses, described or expected to induce similar membrane modifications (eg coronaviruses, flaviviruses, etc.).

## Introduction

Viruses are obligate intracellular parasites that rely on the host cellular machinery to achieve their replication cycle and transmission. Efficient viral infections have evolved from strategies that hijack cell structures, pathways and mediators seeking to counteract these infections. This pro-viral balance sometimes results in dramatic intracellular endomembrane alterations [1,2]. One striking example of viral infection impact on organelles is the plasticity of mitochondrial morphodynamics. Indeed, while some viruses like human Cytomegalovirus or Hepatitis B virus induce mitochondrial fission [3,4], other viruses like Dengue or Sendai trigger mitochondrial network [5,6]. Mitochondrial antiviral signaling (MAVS) is an adaptor protein, localized at outer mitochondria membranes and at endoplasmic reticulum (ER)-mitochondria tethering interfaces, that makes the connection between mitochondria and innate immunity machinery [7]. In this context, MAVS translates the detection of viral RNA by RIG-I Like Receptors (RLR) into an anti-viral type I interferon response [8–11]. Interestingly, activation of RLR induces elongation of the mitochondrial network [5,6] illustrating the functional interplay between mitochondrial morphodynamics regulation and the molecular control of innate immunity in host cells. Conversely, artificial elongation of mitochondria by depletion of DRP1, one major effector of the mitochondrial fission machinery, has been described to decrease the expression of interferon β in the context of Dengue virus infection [5]. A recent study presents a mechanistic relation between mitochondrial membranes dynamics and innate immunity [12]. Following RNA sensing through RIG-I like receptors, the TBK1 kinase directly and massively phosphorylates DRP1 at S412, S684. These phosphorylation events prevent DRP1 oligomerization and aggregation to the mitochondria and thereby abrogate the mitochondrial fission activity.

Influenza virus genomic RNA is also sensed by mitochondrial RIG-I [13]. However, cell endomembranes alterations during the course of influenza virus infection have not been extensively explored. Briefly, influenza is a major cause of respiratory diseases and is responsible for the death of more than half million people worldwide each year [14]. The current therapeutic arsenal associated with influenza is restricted to two viral targets, M2 ion channel and neuraminidase proteins, with the recent emergence of viral PB2 protein as a third viral target [15]. The main drawback associated with these direct-acting antiviral compounds is the rapid emergence of viral strains resistant to treatments an observation that underscore the importance of examining host directed strategies in the fight against Influenza virus infection [16]. One approach is to study the mechanisms by which the virus induces intracellular endomembrane alterations. In host cells, influenza infection is described to induce autophagy [17] and fragmentation/dispersal of the Golgi apparatus [18]. Moreover, when addressed to the

mitochondria, the viral protein PB1-F2 induces mitochondrial network fragmentation [19]. Classical *in vitro* experimental settings used in influenza virus infection studies routinely include serum deprivation and are thus expected to artificially affect endomembranes network and function, a situation that could mask viral induced phenotypes. In this study, we analyzed the host cellular endomembranes morphodynamics alterations associated with influenza infection in optimized and stress reduced experimental conditions. We report for the first time the unexpected elongation of the mitochondrial network and perturbation of endoplasmic reticulum-mitochondria contact sites as a direct consequence of influenza virus infection and we demonstrate that influenza RNA, a RIG-I ligand, is sufficient to induce this phenotype. We show that influenza infection shifts the fusion/fission molecular machinery towards a fusion state, a situation that promotes the observed mitochondrial network elongation. Finally, we provide evidence that this elongated phenotype can be targeted in a RIG-I pathway dependent manner, with a pro-fission molecule to inhibit influenza virus replication.

## Results

### Influenza virus infection induces mitochondria elongation and alters mitochondria-ER contact sites

Confocal fluorescence microscopy analyses of human alveolar epithelial A549 cells infected by H1N1 influenza A virus was performed to gain insights into subcellular organellar alterations possibly induced by influenza virus infection. In order to minimize additional cellular stress associated with infection, we set up a protocol with reduced cell exposure to trypsin and serum deprivation, conditions often associated with viral infection *in vitro*. Indeed, trypsin treatment profoundly modifies cellular morphology and distribution of viral components (S1A Fig) and serum deprivation induces mitochondria and Golgi fragmentation (S1B Fig). In our experimental design, cells were exposed to influenza virus for 1h in presence of trypsin in serum free medium before incubation in complete medium. The kinetics of trafficking and subcellular localizations of viral M2 and NP proteins were not modified in these conditions (S1C, S1D and S1E Fig).

As expected, H1N1 influenza A infection induced accumulation of LC3-positive punctate staining in A549 cells (S2A Fig), suggesting an already reported autophagy induction [20] and staining with GM130, a Golgi marker, revealed fragmentation and dispersion of the Golgi apparatus [18] (S2B Fig). While the number and distribution of late endosomes (LAMP1 punctate staining) was not affected, EEA1 labeling revealed significantly less early endosomes following viral infection (S2C and S2D Fig) and GFP-tagged Sec61β, used as a marker of ER morphology, remained similarly distributed in infected and uninfected cells (S2E Fig) suggesting that no major ER morphology changes occur during H1N1 infection. Unexpectedly, while uninfected A549 cells exhibited globular mitochondria, H1N1 infection clearly induced an elongated mitochondria phenotype as determined by TOMM20 immunofluorescence analysis (Fig 1A and 1B).

Influenza viruses are sensed by different pattern-recognition receptors inside cells leading to activation of the innate immune system. The main cytosolic receptor is the RIG-I protein, which induces type I interferon production through the mitochondrial protein MAVS [9–11,21]. Given the importance of MAVS in the regulation of the RIG-I pathway, we focused on the characterization of the mitochondria morphodynamics phenotype in H1N1 infected cells. Electron microscopy analysis confirmed a statistically significant increase of mitochondria length (Fig 1C and 1D) thus excluding a potential accumulation of inter-mitochondrial junctions [22]. The mitochondrial elongation was detectable at 280 min post-infection and reached its maximum at 520 min, the latest time point measured of our time-lapse analysis (Fig 1E).

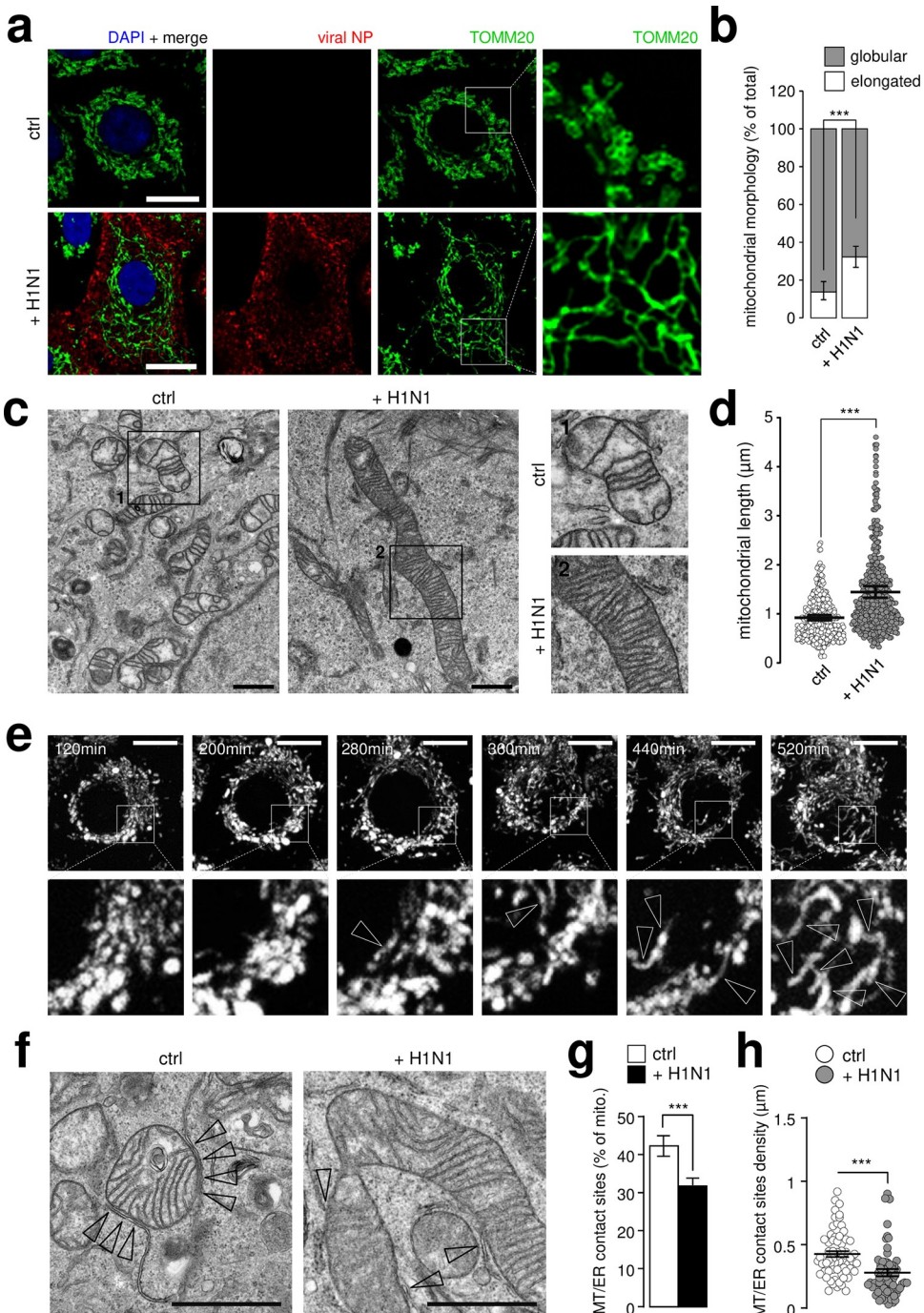

**Fig 1. Influenza virus infection induces mitochondria elongation and alters ER-mitochondria contact sites. a,**
A549 cells infected with H1N1 influenza A virus at MOI 1 for 24h (or mock infected) were immunostained with anti-
TOMM20 antibody (green), anti-NP antibody (red) and DAPI (blue); cropped areas show mitochondria morphology,
elongated in infected condition. **b,** Quantification of elongated and globular mitochondria (TOMM20 signal) from
single cells illustrated in (a) (N = 150 cells from three independent experiments). **c,** Electron micrographs (EM) from
A549 cells infected (or not) with influenza A H1N1 virus at MOI 1 for 24h; cropped areas show the mitochondria
morphology, hyper-fused in the infected condition. **d,** Quantification of mitochondrial length in EM images from
A549 cells illustrated in c (N = 250 mitochondria, from three independent experiments). **e,** Mito-tracker time lapse
video-microscopy on A549 cells infected with H1N1 virus from 120min to 520min post-infection; arrowheads show
the formation of elongated mitochondria network. **f,** EM pictures from infected (or not) A549 cells with influenza A
H1N1 virus at MOI 1 for 24h; arrowheads show mitochondria-ER contact sites. **g,** Quantification of mitochondria-ER

contact sites in EM images from A549 cells illustrated in f (N = 100 mitochondria, from three independent experiments). **h**, Quantification of mitochondria-ER contact sites density in EM images from A549 cells illustrated in f (N = 100 mitochondria, from three independent experiments). All scale bars = 10μm, except in EM (1 μm). For evaluating significance of differences observed in b,d,g and h, a two-tailed Student's *t* test was used (*** indicates p<0.0001).

Mitochondria-ER contact sites membrane domains are proposed to mediate MAVS-dependent innate immune signaling [7]. By a thorough analysis of electron microscopy pictures, we also provide evidence that mitochondrial elongation is associated with a significant reduction of contact sites number and density following infection (Fig 1F, 1G and 1D). Overall, our analyses suggest that intracellular organelles are profoundly modified upon H1N1 virus infection with striking elongation of the mitochondrial network.

Interestingly, similar observation was made with influenza A H3N2 subtype and the more distant influenza B strain (IVB) suggesting that mitochondrial elongation is broadly associated with influenza viruses infection (S3A Fig). Moreover, in addition to A549 cells, influenza A H1N1, H3N2 and influenza B viruses also elicited mitochondria elongation in MDCK renal epithelial cells (S3B Fig). Finally, influenza virus also induced this phenotype in HEK293T cells (S4A and S4B Fig) arguing for a general hallmark associated with influenza infections.

In the context of antiviral response studies, it has been reported that synthetic double stranded RNA poly I:C, known to bind and activate RIG-Like Receptors, also induces mitochondria elongation [6]. To test if H1N1 influenza A RNA is one of the viral elements responsible for this phenotype, we transfected A549 cells with a synthetic 89-mer 5'triphosphate hairpin RNA (VhpRNA) derived from the influenza A H1N1 genome. This hairpin RNA is known to be a RIG-I ligand that when introduced in cells triggers a potent interferon response [23]. In cells transfected with the influenza A VhpRNA, the elongated mitochondrial phenotype that we observed in virus infected cells (see Fig 1) appeared 6h hours post-transfection and was concomitant with the detection of type I beta interferon production (Figs 2A and 2B and S5A). Accordingly, type I interferon secretion was maximal when the highest number of cells had elongated mitochondria (S5A Fig). These data indicate that influenza A RNA is sufficient to induce mitochondria elongation in host cells. Strikingly, Inarigivir, a RIG-I chemical ligand and activator of the RIG-I pathway, did not reproduce the mitochondria elongation we observed with VhpRNA (S5B and S5C Fig), suggesting that this phenotype is associated with natural RIG-I ligands specificity. Like for the virus, we also explored the modification of ER-mitochondria contact sites following VhpRNA transfection. We showed a significant reduction of contact sites number and density in electron microscopy (Fig 2C, 2D and 2E). The reduction was validated in confocal microscopy experiment where PTPIP51 presence (a contact site resident protein, [24]) was significantly reduced at the ER/mitochondria interface (Fig 2F and 2G).

To further investigate the regulation of the molecular machinery involved in mitochondria alteration promoted by viral infection, we examined the subcellular localization of DRP1 (Dynamin-1-like protein, also referred as DNM1L), the main protein involved in the mitochondrial fission machinery. Interestingly, we observed that viral infection induced an important induced a decrease in DRP1 presence at mitochondria (Fig 3A and 3B). Conversely, infection increased OPA1 signal at the mitochondria, suggesting a coalescence of the protein at this compartment, OPA1 being a key protein in mitochondrial fusion [25] (Fig 3C and 3D). This phenomenon was accompanied by a slight decrease in DRP1 protein total amount (S6 Fig). It is also associated with a decrease expression of MFN2 and MFN1 proteins (S6 Fig). MFN2 is described as acting like a tether between ER and mitochondria compartments in homotypic interactions and heterocomplexes with MFN1 [26]. We did not detect any change

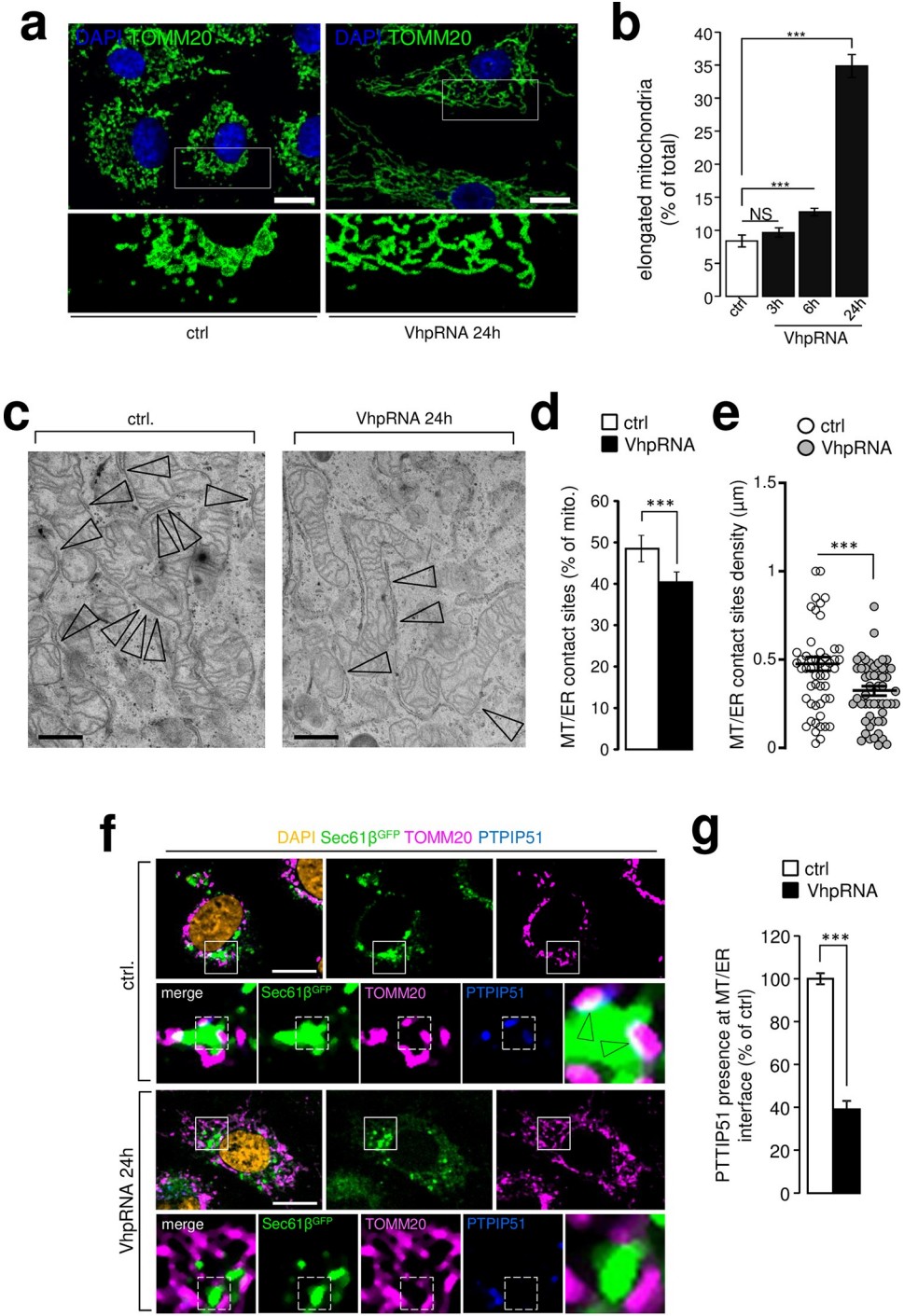

**Fig 2. Expression of H1N1 viral hairpin RNA recapitulates mitochondria membrane modifications induced by the virus. a**, A549 cells were transfected (or not) with a 89-mer 5'triphosphate hairpin RNA synthesized from a sequence of influenza A H1N1 virus genome (viral hairpin RNA, VhpRNA) and immunostained with anti-TOMM20 antibody (green) and DAPI (blue); cropped areas show mitochondria morphology, elongated in transfected condition (N = 3). **b**, Quantification of elongated mitochondria illustrated in (a) at 3h, 6h and 24h (N = 3). **c**, EM pictures from VhpRNA transfected (or not) A549 cells, 24 post-transfection; arrowheads show mitochondria-ER contact sites. **d**, Quantification of mitochondria-ER contact sites in EM micrographs from A549 cells illustrated in c (N = 100 mitochondria, from three independent experiments). **e**, Quantification of mitochondria-ER contact sites density in EM images from A549 cells illustrated in c (N = 100 mitochondria, from three independent experiments). **f**, A549 cells, expressing SEC61β-GFP, transfected (or mock transfected) with VhpRNA were immune-stained with anti-TOMM20

antibody (pink), anti-PTPIP51 antibody (blue) and DAPI (blue); cropped areas show PTPIP51 presence at ER-mitochondria contact sites, and absence in transfected conditions. **g**, Quantification of PTPIP51 presence at mitochondria-ER contact sites A549 cells illustrated in d (N = 100 mitochondria, from three independent experiments For evaluating significance of differences observed in b, d, e and g, a two-tailed Student's *t* test was used (\*\*\* indicates p<0.0001).

in OPA1 isoforms expression following infection even if OPA1 total amount is decreased (S6 Fig). The long isoform, previously associated with fusion [27], remains the main isoform. These data are consistent with a reduction of ER-mitochondria contact sites in the context of mitochondrial fusion induced by influenza virus infection. Interestingly, DRP1 expression at the mitochondria is also slightly decreased following VhpRNA transfection, associated with a total decreased expression of the protein (S5D–S5G Fig). Altogether these data indicate that

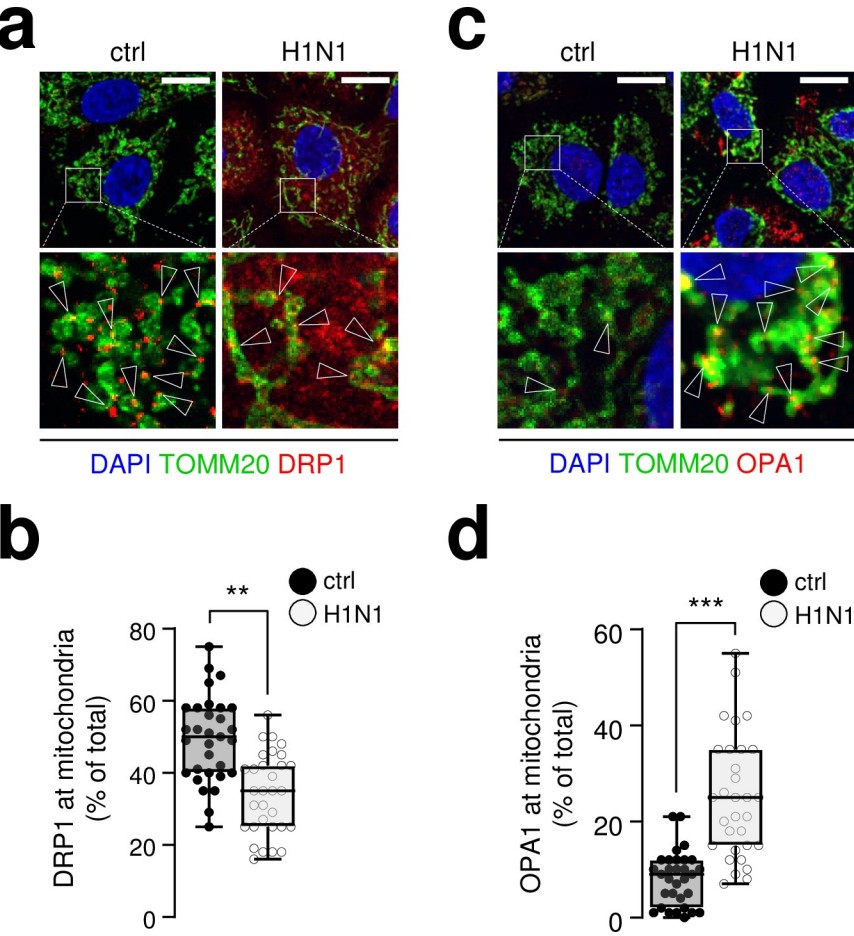

**Fig 3. Influenza infection shifts the fusion/fission molecular machinery balance towards a fusion state. a**, A549 cells infected (or mock infected) with influenza A H1N1 virus at MOI 1 for 24h were immunostained with anti-TOMM20 antibody (green), anti-DRP1 antibody (red) and DAPI (blue); arrowheads indicate recruitment of DRP1 onto the mitochondrial surface (TOMM20). **b**, Quantification of DRP1 signal on TOMM20 positive structures from A549 cells illustrated in (a) (N = 30 cells from three independent experiments). **c**, A549 cells infected (or not) with influenza A H1N1 virus at MOI 1 for 24h were immunostained with anti-TOMM20 antibody (green), anti-OPA1 antibody (red) and DAPI (blue); arrowheads indicate accumulation of OPA1 at the mitochondrial matrix (TOMM20). **d**, Quantification of OPA1 signal on TOMM20 positive structures from A549 cells illustrated in (e) (N = 30 cells from three independent experiments). Scale bars = 10μm. For evaluating significance of differences observed in b, d and f, a two-tailed Student's *t* test was used (\*\*\* indicates p<0.0001).

the viral hairpin RNA works as a surrogate of viral infection on the main mitochondrial phenotypes explored in this study.

## Drug designed mitochondria fragmentation inducer (Mito-C) inhibits influenza replication

In the context of H1N1 viral infection, we thought to evaluate whether the mitochondrial fusion phenotype we just described was required for influenza pathogenesis. We therefore investigated the effects of Mito-C, a molecule recently developed by ENYO Pharma and previously described to induce mitochondria fragmentation on virus replication [24]. Our hypothesis was that Mito-C could counteract the effects of influenza virus infection on mitochondria fusion, restore a normal phenotype and inhibit influenza replication. First of all, cell viability assays demonstrated that Mito-C was not toxic on A549 cells (S7 Fig). To detect putative antiviral effect of Mito-C, A549 cells were then infected by H1N1 influenza A virus at a MOI (multiplicity of infection) of 0.1 and virus associated neuraminidase activity (NA) was quantified in the infected cells culture supernatant 2 days post-infection. Mito-C induced a dose-dependent inhibition of NA activity with an IC50 of ±50nM comparable to oseltamivir, the most efficient influenza antiviral compound also active in the low nanomolar range of concentration (Fig 4A). Consistent with these data, a growth kinetics study confirmed the inhibition of viral replication at each time point between 24h and 96h post infection (S8 Fig). This inhibitory effect was further validated in a plaque forming unit (PFU) assay which revealed that Mito-C exhibited a 10-fold inhibition of the virus titer at 2μM (Fig 4B and 4C). Similarly, viral NP expression in western-blot analysis was shown to be dramatically decreased upon Mito-C treatment (Fig 4D and 4E).

We next evaluated the mitochondria phenotype in influenza virus infected cells treated or not with Mito-C. Interestingly, while H1N1 infection promoted mitochondrial elongation as shown (Fig 1A and 1C), the treatment with Mito-C reverted this phenotype as assessed by fluorescence (Fig 4F and 4G) and electron microscopy analyses (Fig 4H and 4I). At the sub-organelle level, we further show a restoration of mitochondria-ER contact sites integrity following Mito-C treatment in infectious conditions (Fig 4J, 4K and 4L). Both the percentage of mitochondria engaged in mitochondria-ER contact sites and the contact sites density were similar to uninfected conditions. As expected, Mito-C also inhibited mitochondria elongation induced by the viral hairpin RNA (S9A and S9B Fig).

## Mito-C potentiates innate immune response in infected cells in a RIG-I dependent manner

The increased number of mitochondria-ER contact sites induced by Mito-C suggests that this compound may activate the innate immune response associated with anti-viral response. Hence, we assessed the impact of the molecule on endogenous interferon λ1 and β expression by RT-qPCR since both cytokines are directly associated with anti-viral RIG-I pathway [28]. Importantly, Mito-C had no effect upon either cytokine's expression in uninfected cells (Fig 5A and 5B). As expected H1N1 viral infection enhanced IFNλ1 and IFNβ expression (Fig 5A and 5B) and remarkably, the cytokine inductions were significantly enhanced when H1N1 infected cells were treated with Mito-C (Fig 5A and 5B). This enhancement is also measured with Mito-C treatment when cells are transfected with the viral hairpin RNA (S9C and S9D Fig). These data are consistent with the observed inhibition of H1N1 virus replication by Mito-C (Fig 4) and suggests that the anti-viral activity of Mito-C activity is dependent on the RIG-I pathway. Indeed, knockout of RIG-I by CRISPR/Cas9 technology (Fig 5C) abrogated the antiviral activity of Mito-C as assessed by neuraminidase activity in treated and untreated

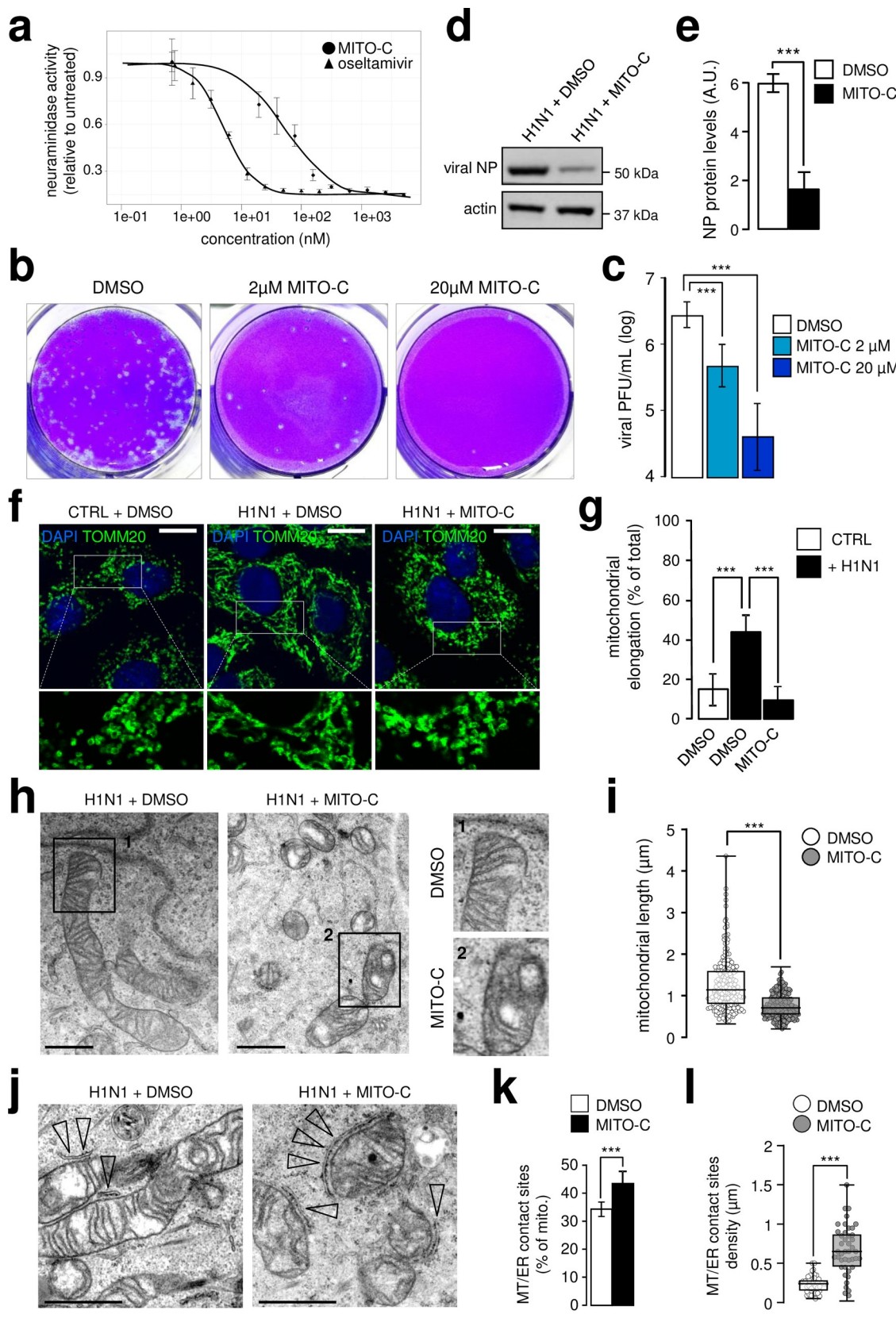

**Fig 4. Drug designed mitochondria fragmentation inducer (Mito-C) inhibits influenza replication. a**, Determination of neuraminidase activity in the supernatant of A549 cells infected with influenza A H1N1 virus at MOI 0.1 for 48h, treated with Mito-C compound or oseltamivir at indicated concentrations (N = 3). **b**, PFU assay in MDCK cells from A549 infected supernatants (MOI 0.1, 48h) treated with Mito-C or DMSO (vehicle) at indicated concentrations. **c**, Quantification of PFU illustrated in (b) (N = 3). **d**, Western blot analysis of NP viral protein in A549 cells infected with H1N1 at MOI 0.1 for 48h, treated with Mito-C at 2µM or DMSO (vehicle). **e**, Quantification of NP western blot showed in (d) (N = 3). **f**, A549 cells infected or not, with H1N1 at MOI 1 for 24h and treated with Mito-C at 2µM or DMSO were immunostained with anti-TOMM20 antibody (green) and DAPI (blue). **g**, Quantification of mitochondrial elongation (TOMM20 signal) from single cells illustrated in (f) (N = 50 cells from three independent experiments). **h**, Electron micrographs (EM) from infected A549 cells with influenza A H1N1 virus at MOI 1 for 24h treated with Mito-C at 2µM or DMSO (vehicle); cropped areas show mitochondria morphology. **i**, Quantification of mitochondrial length in EM images from A549 cells illustrated in (h) (N = 150). **j**, EM pictures from infected A549 cells with influenza A H1N1 at MOI 1 for 24h treated with Mito-C at 2µM or DMSO; arrowheads point at mitochondria-ER contact sites, increased upon Mito-C treatment. **k**, Quantification of mitochondria-ER contact sites number in EM images from A549 cells illustrated in (j) (N = 250 mitochondria, from three independent experiments). **l**, Quantification of mitochondria-ER contact sites density in EM images from A549 cells illustrated in (j) (N = 250 mitochondria, from three independent experiments). All scale bars = 10µm, except in EM (1µm). For evaluating significance of differences observed in c, e, g, i, k and l, two-tailed Student's *t* test was used (*** indicates p<0.0001).

infected cell supernatants (Fig 5D). Overall, these data indicate that the antiviral activity of Mito-C relies on mitochondrial RIG-I pathway engagement and enhancement of interferon responses to influenza A RNA.

## Discussion

We provide here a landscape of cellular endomembrane modifications associated with influenza virus infection. Autophagy induction has been extensively described in the context of influenza infection [17] and Golgi fragmentation is suspected to facilitate the intracellular trafficking of virions [18]. The reduction of the pool of early endosomes has never been reported following infection and further analysis is required to understand the functional impact of this phenotype for the virus and for its host.

Mitochondria elongation in H1N1 infected cells was unexpected as previous reports identified the full length form of the viral protein PB1-F2 as a pro-fission factor for mitochondria [19]. The nature of the virus strain studied appears critical since approximately 11% of H1N1 and 96% of highly pathogenic H5N1 encode for full length PB1-F2. We demonstrate here that the experimental settings also profoundly affect the structure and dynamics of mitochondria. Serum starvation, systematically used in H1N1 related studies to allow multiple infection cycles, induces mitochondria fragmentation. Hence, we chose to reduce the cell exposure to this cellular stress and revealed that H1N1 viral infection induced mitochondrial network elongation.

Conflicting results have been reported about the functional consequences of mitochondria elongation in an infectious context. In the context of Sendai virus infection, elongation induced by DRP1 depletion enhances IFNβ expression [6] while it is repressed in the context of Dengue virus infection [5]. This apparent discrepancy can be reconciled by considering the number of ER-mitochondria contact sites: Sendai infection increases their number and Dengue virus disrupts them. Interestingly, ER-mitochondria contact sites are platforms for MAVS-dependent activation of innate immune response [7]. Here, in agreement with what was previously described for Dengue, influenza virus infection decreases the number of contact sites arguing for an acute and rapid limitation of innate immune response. The enhancement of interferon expression with Mito-C, a pro-fission factor that also restores the number of ER-mitochondria contact sites, supports the hypothesis that mitochondria elongation is detrimental to innate immunity. Our data would indicate that, like for HCV, the ER-Mitochondria contact sites resident MAVS mediates the RIG-I signaling pathway in response to influenza virus infection. As a consequence of ER-mitochondria contact sites disruption,

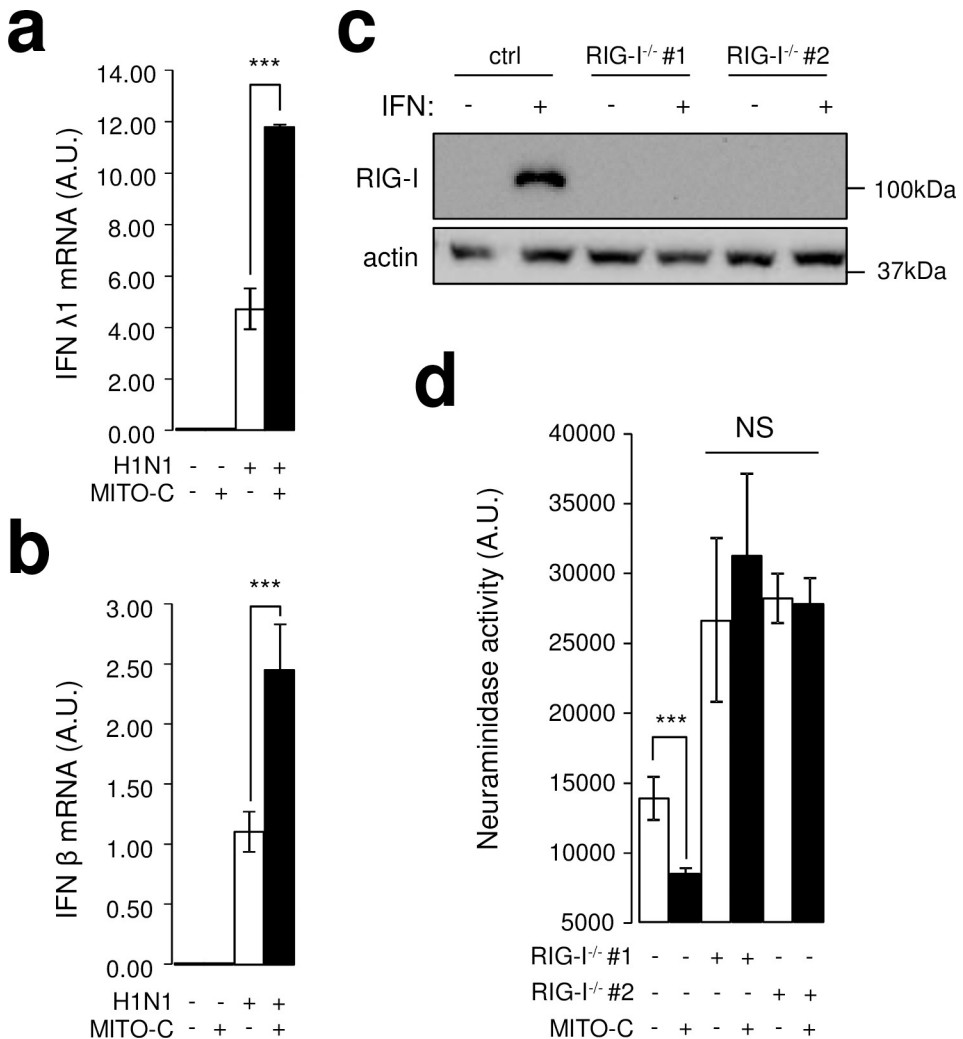

**Fig 5. Mito-C potentiates innate immune response in infected cells in a RIG-I dependent manner. a**, RT-qPCR analysis of IFNλ1 mRNA from A549 cells infected (or mock infected) with H1N1 at MOI 1 for 24h and treated with Mito-C at 2μM or with DMSO (N = 3). **b**, RT-qPCR analysis of IFNβ mRNA from A549 cells infected (or mock infected) with H1N1 at MOI 1 for 24h and treated with Mito-C at 2μM or with DMSO (N = 3). **c**, Western blot analysis of RIG-I expression in two independent A549 RIG-I$^{-/-}$ cell lines with and without IFNα treatment (CRISPR/Cas9, guide #1 and guide #2) **d**, Determination of neuraminidase activity in the supernatant of A549 wild type and RIG-I$^{-/-}$ cells (guide #1 and guide #2), infected with H1N1 at MOI 1 for 24h, treated with Mito-C at 2μM or DMSO (N = 3). For evaluating significance of differences observed in c, e, g, I, k and l two-tailed Student's $t$ test was used (*** indicates p<0.0001, NS for non-significant).

MAVS should thus be redistributed in other compartments. Alternatively, this disruption could also alter the assembly of MAVS signalosome (i.e. MAVS aggregation) and membrane recruitment of TRAF3 or TRAF6 [7].

Finally, the identification of influenza RNA as a sufficient viral element to induce mitochondria elongation is consistent with what has been reported previously [6,12]. In addition, we provide evidence that the viral RNA also recapitulates the impact of the virus on ER-mitochondria contact sites density and on the stability of DRP1 on the mitochondria. This indicates that the viral RNA is a major driver of the mitochondrial membranes morphodynamics alterations occurring during influenza virus infection and can be considered as a surrogate of the virus in this process. The inability of Inarigivir to elicit this phenotype is quite unexpected but

suggests that RIG-I engagement is not sufficient to induce mitochondria elongation. Obviously, the nature of RIG-I ligands is different, and this might have a differential impact on RIG-I conformation and/or protein interactions. Following the Chen et al. study [12], it will be interesting in the future to test the recruitment of DRP1 to the MAVS signalosome, its interaction with MAVS and its phosphorylation status following Inarigivir binding to RIG-I versus VhpRNA binding to RIG-I.

Our data also supports the idea that any viral RNA could be sufficient to induce mitochondria elongation. In this context, it seems that Mito-C antiviral capability relies on RIG-I engagement as a consequence of influenza infection for the pro-interferon activity that we report here. Indeed, no anti-viral activity is detectable in RIG-I knock-out cells. Together with its anti-Dengue virus activity, this finally suggests that Mito-C compound could have a broad-spectrum activity against viruses that are sensed by RIG-I and that induce mitochondria elongation [12, 24]. In the current epidemic context, it is tempting to propose the repositioning of Mito-C against SARS-CoV-2, since SARS-coronavirus ORF9b protein was recently shown to induce mitochondrial network elongation and to inactivate RLR pathway by degrading MAVS signalosome [29]. Further studies will be required to understand in detail the mechanisms by which influenza related viruses provoke alteration of mitochondria morphodynamics, but our study highlights the importance of the interconnection between innate immunity machinery regulation and mitochondria morphology in a viral context.

## Materials and methods

### Cell lines and viruses

Human lung adenocarcinoma (A549, ATCC), Human Kidney (293T, ATCC) and Madin-Darby Canine Kidney (MDCK, ATCC) cell lines were grown in Dulbecco's modified Eagle's medium (DMEM, Gibco) supplemented with 10% heat-inactivated fetal bovine serum (FBS, Pan Biotech) and 50 IU/ml penicillin G, 50 mg/ml streptomycin (Gibco), at 37˚C under 5% $CO_2$. A549-Sec61β-GFP stable cell line was obtained after plasmid transfection (Addgene, 15108) using Fugene HD transfection Reagent (Promega, #E2311) following the manufacturer's instructions. 48h after transfection, transfected cells were submitted to selection by geneticin treatment (Themo Fisher, #0131035) at 1400μg/ml for 10 days. For CRISPR–Cas9-mediated RIG-I (DDX58) gene disruption, A549 cells stably expressing Cas9 were generated as described before [30]. Briefly, Cas9-expressing A549 cells were transduced with control or RIG-I guide RNA expressing Lentiguide-Neo vectors and selected with G418 (1mg/ml) for at least 15 days. The guide RNA coding sequences used were as follows: g1-CTRL 5' -AGCACG TAATGTCCGTGGAT, g2-CTRL 5′ -CAATCGGCGACGTTTTAAAT, g1-RIG-I 5' -GGGT CTTCCGGATATAATCC, g2-RIG-I 5'–TTGCAGGCTGCGTCGCTGCT. RIG-I CRISPR-Cas9 gene disruption was confirmed by western-blotting following IFNα treatment (INTRON A; Merck Sharp & Dohme, 1,000 U/ml) for 24 h. Influenza A virus A/New Caledonia/2006 (H1N1) (clinical isolate), influenza A virus A/H3N2/ Wyoming/2002 and Influenza B virus B/ Virginia/2009 (ATCC 3VR-1784) were propagated in MDCK cells.

### Infection experiments and Mito-C treatment

Two different experimental designs were used to carry out influenza infection experiments. To perform multi-round infection, A549 cells were infected with influenza A H1N1 virus at a multiplicity of infection (MOI) of 0.1 in infection medium (DMEM serum-free medium supplemented with 50 IU/ml penicillin, 50 mg/ml streptomycin and 0.2 mg/ml TPCK-trypsin (Sigma, #4352157)). After 48h, supernatants were harvested to test neuraminidase activity (Munana test) and viral titer (PFU). Cells were harvested as well to test viral protein levels in

western blot. To perform single-round infection, A549 or MDCK cells were infected with the indicated influenza virus at MOI of 1 in infection medium. After 1h, cells were washed and the medium was replaced with warm DMEM medium supplemented with 10% heat inactivated FBS and 50 IU/ml penicillin G, 50 mg/ml streptomycin. At indicated times, cells were lysed to carry out western blot assays or fixed in 4% PFA to perform immunofluorescence assays. When indicated, Mito-C was added to A549 cells at the same time as influenza virus at the indicated concentrations. As negative control, the same amount of DMSO was used.

## MitoC treatment and 3p-hpRNA/LyoVec transfection

A549 cells were plated in 24 wells plates, on coverslips for imaging, and, the following day treated with 2 μM Mito-C during 30 min at 37˚C. After the treatment, cells were transfected with 3p-hpRNA/LyoVec (tlrl-hprnalv InvivoGen) during 30 min, 6 h and 24 h at 37˚C and prepared for RT-PCR and immune-fluorescence analyses.

## Neuraminidase assay

A fluorometric assay was used to detect influenza virus neuraminidase activity. This viral enzyme is able to cleave the 2′-(4-Methylumbelliferyl)-α-D-N-acetylneuraminic acid sodium salt hydrate (Munana, Sigma, #M8639), generating a fluorescent product that can thus be quantified. In 96-black plate, 25μl of infection supernatants were added to 75μl of 20 mM Munana in PBS containing $Ca^{2+}$ and $Mg^{2+}$. After 1h incubation at 37˚C, the reaction was stopped by adding 100μl of glycine 0.1 M and ethanol 25%. Fluorescence was recorded using TECAN Spark 20M at wavelengths of 365nm for excitation and 450nm for emission. The R statistical software was used to perform the dose-response analysis (drc package) and plot the dose-response data and the associated logistic regression (ggplot2 library).

## Plaque forming unit assay (PFU)

PFU assays were used to titer viral stocks and to determinate titer of Mito-C antiviral effect in infected cells. Briefly, supernatants from these two experiments were serial diluted following 1:3 dilution factor in infection medium. 500μL of each virus dilution was added to a monolayer of MDCK cells in 6-well plates. Cells were incubated 90 min for viral uptake and then virus was washed off. A mix of 2% SeaKem agarose (Lonza, #50011) and 2xMEM (Gibco) supplemented with 0.4μg/ml TPCK trypsin (1:1) were added to each well. Plates were incubated for 72 h at 37˚C, 5%$CO_2$. Cells were fixed in a 10% PFA solution and then stained with crystal violet solution (0.3% crystal violet, 10% ethanol in water) for 30min at room temperature and abundantly rinsed with distilled water. Infectious titers were calculated considering the corresponding dilution factor after PFU numeration.

## Growth curve and multicycle Influenza infection measurements

For multicycle influenza A infection experiment, confluent A549 cells were infected in a 12 well plate format with A/Victoria/3/75 virus at a MOI of 0.001 PFU/cell and then incubated for 1h at 37˚C and 5% $CO_2$. The inoculum was removed, cells were washed once with 1ml PBS and 1ml of serum-free DMEM with 0.2ug/ml of TPCK-treated trypsin containing either Mito-C (2μM), Oseltamivir (50nM) or DMSO was added before incubating cells at 37˚C. Infections were performed in triplicates. At 24, 48, 72 and 96h post-infection, samples were collected and frozen for viral titration by standard plaque assay on MDCK cells.

## Western blot analyses

Western blot (WB) analyses were performed with precast gradient gels 4–12% Bis-Tris (Invitrogen) and 3–8% Tris-Acetate (for DRP1 and OPA1 detection) using standard methods. Briefly, cells were washed twice with cold PBS and lysed on ice in NP-40 buffer (150 mM NaCl, 20mM Tris-HCl at pH8, 1mM EDTA, 1% NP-40) for 20min in the presence of the complete protease inhibitor cocktail (Roche). The soluble fraction was recovered after centrifugation at 13000g for 20min, then protein concentration was measured by the BCA protein assay kit (Pierce) using bovine serum albumin (BSA) as a standard. Protein lysates were diluted in Laemmli SDS sample buffer (Thermo Fisher) before being incubated for 10 min at 95˚C. After transfer, PVDF membranes were blocked in 5% milk/Tween 0.1% buffer for 1h and incubated with primary antibodies overnight at 4˚ in 5% BSA/TBS—Tween 0.1% buffer. After secondary HRP antibody staining, membranes were visualized using ECL western blotting substrate (Thermo Fisher) and a Pxi Imaging System (Syngene). Quantification of band intensity was carried out using Image J software.

## Immunofluorescence and confocal microscopy

For immunofluorescence analysis (IF), cells were plated on 12-mm glass coverslips and fixed with 4% paraformaldehyde (Sigma) in PBS for 20 min at room temperature. After 3 washes of 10 min each in PBS, cells were blocked with fetal bovine serum (10%) in PBS for 30min. Incubation with primary antibodies was performed in permeabilization buffer (0.05% saponin in 10% FBS-PBS buffer). Coverslips were mounted on microscope slides using prolong diamond antifade mountant with DAPI (Thermo Fisher, #P36962). Images were obtained using a 63x/ 1.4 oil-immersion objective with a ZEISS LSM700 confocal microscope, or a ZEISS LSM880 confocal microscope or a Leica TCS SP5 confocal microscopeAcquisitions were done in sequential mode and fluorescence acquired in separated channels.

## Antibodies and dilutions

Primary antibodies used were the following: mouse anti-TOMM20 (BD Biosciences 612278, 1:200 in IF); rabbit anti-TOMM20 (Abcam ab186734, 1:200 in IF, 1:1000 for WB); mouse anti- DRP1 (BD Biosciences 611112, 1:500 for WB and 1:200 in IF); rabbit anti- PTPIP51 (Novus Biologicals NBP1-84738, 1:200 for IF); rabbit anti- MFN1 (Cell Signaling 14739, 1:500 for WB); rabbit anti-MFN2 (Cell Signaling 9482, 1:500 for WB); rabbit anti- OPA-1 (Cell signaling 67589, 1:500 for WB); mouse anti-GM130 (BD Biosciences 610823, 1:500 in IF); rabbit anti-LAMP1 (Abcam ab24170, 1:500 in IF); mouse anti-EEA1 (BD Biosciences 610456, 1:200 in IF); rabbit anti-LC3 (MBL PM036, 1:200 in IF); mouse anti beta-actin (Sigma A1978; 1:1000 for WB); rabbit anti-influenza A virus M2 protein (Abcam ab5416; 1:1000 for IF); mouse anti-influenza A virus nucleoprotein NP (Abcam ab128193, 1:1000 in IF and WB); mouse anti-influenza B virus nucleoprotein (Abcam ab20711, 1:1000 in IF); RIG-I (Covalab, 102466, 1:1000 in WB). For secondary antibodies, we used for immunofluorescence an Alexa 488 conjugated donkey anti-mouse (Invitrogen #A21202, 1:800), Alexa 488 conjugated donkey anti-rabbit (Invitrogen #A21206, 1:800), Alexa 647 conjugated donkey anti-mouse (Invitrogen #A31571, 1:800), Alexa 647 conjugated donkey anti-rabbit (Invitrogen #A31573, 1:800). For western blotting we used Secondary HRP conjugate anti-rabbit IgG (GE Healthcare, 1:10000) and HRP conjugate anti-mouse IgG (Bio-Rad 1:10000).

## Transmission electron microscopy

Cells were chemically fixed in 2.5% (v/v) glutaraldehyde, 2% (v/v) paraformaldehyde in 0.1 M cacodylate (pH 7.2) buffer for 2h at room temperature, washed in cacodylate, post-fixed with

2% (w/v) osmium tetroxide supplemented with 1.5% (w/v) potassium ferrocyanide (45 min, 4˚C), washed in water, dehydrated in ethanol (increasing concentration from 30 to 100%) and embedded in Epon as described in [31]. Ultrathin sections of cell monolayers were prepared with a Reichert UltracutS ultramicrotome (Leica Microsystems) and contrasted with uranyl acetate and lead citrate. Electron micrographs were acquired on a Tecnai Spirit Electron Microscope (ThermoFischer Scientific) equipped with a 4k CCD camera (EMSIS GmbH, Muenster, Germany) using ITEM software (EMSIS) or using a GATAN Orius 200 camera.

## Live imaging microscopy

A549 cells were seeded in each compartment of Cellview cell culture dishes (4 compartments, glass bottom, Greiner). The following day, cells were infected with influenza A H1N1 virus at MOI of 1 in a final volume of serum-free DMEM of 250 μl. After 1h incubation at 37˚C, the inoculum was removed and 500μl of live microscopy media (DMEM containing no phenol red supplemented with L-glutamine, pyruvate, 10% serum and antibiotics) containing Mito-Tracker Orange CM-$H_2$TMRos (ThermoFisher, dilution 1: 10 000) was added. Uninfected control cells were treated in the same way. Image acquisition was performed using a Nikon inverted microscope (EclipseTi2) coupled with a Dragonfly spinning disk unit (Andor, Oxford Instruments Company). An oil immersion objective (100X Plan Apo lambda, numerical aperture (NA) 1.45 0.13mm) was used. Acquisitions were performed every 5 minutes for 15h (3 fields per condition). A z step of 1μm was used for the z-stacks.

## RNA extraction and qPCR analyses

RNA was extracted from A549 cells using RNeasy mini kit (Qiagen, 74106) according to the manufacturer's instructions. After DNA digestion with Ambion Turbo DNase (Thermo Fisher, "AM2238), 500ng of RNA were subjected to reverse transcription by using the High-Capacity RNA-to-cDNA kit (Thermo Fisher, "4387406). The obtained cDNA was then used for IFNβ and IFNλ1 detection by qPCR with FastStart Universal SYBR green PCR master mix (Roche) using a Light Cycler 480 II (Roche). The following primers were used: IFNβ forward: CGCCGCATTGACCATCTA, IFNβ reverse: GACATTAGCCAGGAGGTTCTC, IFNλ1 forward: GCAGGTTCAAATCTCTGTCACC, IFNλ1 reverse: AAGACAGGAGAGCTGCAAC TC. In experiments related to Mito-C and VhpRNA treatments, total RNA was extracted using the RNeasy kit (Qiagen) employing on-column DNase treatment, according to the manufacturer's instructions. 125 ng cellular RNA were used to generate cDNAs. The cDNAs were then analyzed by qPCR using published TaqMan gene expression assays (Applied Biosystems) for ACTB (Hs99999903_m1), GAPDH (Hs99999905_m1), IFNB1 (Hs01077958_s1) and IFNL1 (Hs00601677_g1). qPCR reactions were performed in triplicate, in universal PCR master mix using the indicated Taqmans. After 10 min at 95˚C, reactions were cycled through 15 s at 95˚C followed by 1 min at 60˚C for 40 repeats. Triplicate reactions were run according to the manufacturer's instructions using a ViiA7 Real Time PCR system (ThermoFisher Scientific). GAPDH and ACTB mRNA expression was used to normalize the samples.

## dsRNA transfection and interferon measurement

A549 cells were transfected with 3p-hpRNA 5' triphosphate hairpin RNA (VhpRNA, 5'pppGG AGCAAAAGCAGGGUGACAAAGACAUAAUGGAUCCAAACACUGUGUCAAGCUUU CAGGUAGAUUGCUUUCUUUGGCAUGUCCGCAAAC- 3'), a 89-mer synthesized from a sequence of influenza A H1N1 virus (Invivogen, 3p-hpRNA) using Lyovec reagent for transfection following the manufacturer's protocol (Invivogen, lyec-12). After 3h, 6h and 24h cells were fixed in PFA 4% for immunofluorescence assays and supernatants were harvested for interferon

beta (IFNβ) quantification. IFNβ measurement was performed using verikine-HS human IFN beta serum ELISA Kit (PBL assay science, 41415–1) following manufacturer's instructions.

## Toxicity assays

A549 cells were treated for 48h at indicated concentrations with Mito-C. As negative control, the same amount of DMSO was used. Two different tests were performed. First, cell death was quantified by a propidium iodide (PI) assay according to the manufacturer's protocol (Thermo Fisher, #P1304MP). Data were analyzed on a BD Accuri C6 Plus flow cytometer (BD Biosciences) and processed using BD Accuri C6 software (BD Biosciences). Second, levels of lactate dehydrogenase (LDH) in supernatants were quantified using the cytotoxicity detection kit (Roche, #11644793001) following manufacturer's instructions. Absorbance was recorded using TECAN Spark 20M at wavelengths of 496nm.

## Image analysis and statistics

Image analysis was performed using Image J, ICY and Zeiss ZEN softwares. For mounting representative images, background was reduced using brightness and contrast adjustments applied to the whole image. For analyzing the recruitment of DRP1 on mitochondria, a mask on the fluorescence in TOMM20 channel was used to define the region to measure the total intensity of the DRP1 signal in the DRP1 channel. For statistics of data we used GraphPad Prism software. After evaluation of mean, standard deviation and standard errors we evaluate data distributions using normality test: KS normality test, D'Agostino & Pearson omnibus normality test and Shapiro-Wilk normality test. Data where processed as normally distributed when at least 2 out of 3 tests resulted positive, otherwise they were processed as non-parametric distributions. Gaussian distributions were analyzed with the more appropriate t-test or ANOVA and non-Gaussian distributed sets of data were evaluated with the more appropriate non-parametric test, Mann-Whitney or ANOVA, as specified in figure legends.

## Supporting information

**S1 Fig. Experimental set-up for viral infections. a**, A549 cells infected with influenza H1N1 virus at MOI 1 for 24h without serum and treated with trypsin (or not) were immunostained with anti-NP antibody (green) and DAPI (blue). **b**, A549 cells cultured with serum (or not) for 24h were immunostained with anti-TOMM20 antibody (green), DAPI (blue, left panel) and anti-GM130 antibody (green) and DAPI (blue, right panel); cropped areas show mitochondria and Golgi morphology, modified upon serum starvation. **c**, A549 cells were infected with H1N1 virus at MOI 1 in serum-free medium supplemented with trypsin. After 1h, cells were washed, and the medium was replaced (or not) with medium containing serum. Cells were immunostained after 6h, 14h and 24h post infection with anti-NP antibody (green), anti-M2 antibody (red) and DAPI (blue). **d**, Quantification of infected cells (NP signal) from single cells illustrated in (c). **e**, Quantification of infected cells (M2 signal) from single cells illustrated in (c). Scale bars = 10μm. For evaluating significance of differences observed in d and e two-tailed Student's *t* test was used (** indicates p<0.001, NS for non-significant). (TIF)

**S2 Fig. Organellar alterations induced by influenza A H1N1 infection. a**, A549 cells infected (or not) with H1N1 virus at MOI 1 for 24h were immunostained with anti-LC3 antibody (green) and DAPI (blue) and pictures were quantified for LC3 positive structures per cell (N = 30 cells from three independent experiments). **b**, A549 cells infected (or not) with H1N1 virus at MOI 1 for 24h were immunostained with anti-GM130 antibody (green) and DAPI (blue)

and pictures were quantified for fragmented Golgi from single cells (N = 30 cells from three independent experiments). **c**, A549 cells infected (or not) with H1N1 virus at MOI 1 for 24h were immunostained with anti-LAMP1 antibody (green) and DAPI (blue) and pictures were quantified for LAMP1 positive structures per cell (N = 30 cells from three independent experiments). **d**, A549 cells infected (or not) with H1N1 virus at MOI 1 for 24h were immunostained with anti-EEA1 antibody (green) and DAPI (blue) and pictures were quantified for EEA1 positive structure per cell (N = 30 cells from three independent experiments). **e**, A549-Sec61β-GFP stable cell line was infected (or not) with H1N1 virus at MOI 1 for 24h and immunostained with DAPI (blue); cropped areas show ER morphology (N = 30 cells from three independent experiments). Scale bars = 10μm. For evaluating significance of differences observed in a, b, c and d, a two-tailed Student's *t* test was used (*** indicates p<0.0001, NS for non-significant).
(TIF)

**S3 Fig. Mitochondria elongation is not a unique signature of influenza A H1N1 viruses. a**, A549 cells infected (or not) with influenza A H1N1, influenza A H3N2 or influenza B (IBV) viruses at MOI 1 for 24h were immunostained with anti-TOMM20 antibody (green), anti-NP antibody (red) and DAPI (blue) and pictures were quantified for mitochondrial elongation from single cells (N = 50 cells from three independent experiments); cropped areas show mitochondria morphology. **b**, MDCK cells infected (or not) with influenza A H1N1, influenza A H3N2 or IBV viruses at MOI 1 for 24h were immunostained with anti-TOMM20 antibody (green), anti-NP antibody (red) and DAPI (blue) and pictures were quantified for mitochondrial elongation from single cells (N = 50 cells from three independent experiments); cropped areas show mitochondria morphology; cropped areas show mitochondria morphology. Scale bars = 10μm. For evaluating significance of differences observed in a and b two-tailed Student's *t* test was used (*** indicates p<0.0001; NS for non-significant).
(TIF)

**S4 Fig. H1N1 induces mitochondria elongation in HEK293T cells. a**, HEK293T cells infected (or not) with influenza A H1N1 at MOI 1 for 15h were immunostained with anti-TOMM20 antibody (green), anti-NP antibody (red) and DAPI (blue) **b**, Pictures were quantified for mitochondrial elongation from single cells (N = 50 cells from three independent experiments); cropped areas show mitochondria morphology. Scale bars = 10μm. For evaluating significance of differences observed in a and b two-tailed Student's *t* test was used (*** indicates p<0.0001; NS for non-significant).
(TIF)

**S5 Fig. Expression of H1N1 viral hairpin RNA is sufficient to induce mitochondria elongation. a**, A549 cells were transfected (or not) by VhpRNA for 3h, 6h and 24h and IFNβ secretion was measured in cell supernatant (N = 3). **b**, A549 cells were treated (or not) by 1μM or 10μM Inarigivir for 6h and 24h and immunostained with anti-TOMM20 antibody (green) and DAPI. (blue). Cropped areas show mitochondria morphology. Scale bars = 10μm. For evaluating significance of differences observed in a and b two-tailed Student's *t* test was used (*** indicates p<0.0001; NS for non-significant).**c**, Pictures exemplified in b were quantified for mitochondrial elongation from single cells (N = 50 cells from three independent experiments). **d**, Representative western blot analysis of DRP1 in A549 cells transfected (or not), with VhpRNA, 24h post-transfection. **e**, Quantification of DRP1 western blots as showed in (c) (N = 4). **f**, A549 cells transfected (or mock transfected) with VhpRNA were immune-stained with anti-DRP1 antibody (green), anti-TOMM20 antibody (red) and DAPI (blue); cropped areas show DRP1 at the mitochondria (N = 3). **g**, Quantification of DRP1 signal on TOMM20 positive structures from A549 cells illustrated in (i) (N = 30 cells from three independent

experiments). For evaluating significance of differences observed in a, c, e and g, a two-tailed Student's *t* test was used (*** indicates p<0.0001).
(TIF)

**S6 Fig. Expression of the proteins of the fusion/fission machinery following infection. a**, Representative western blot analysis of DRP1, OPA1, MFN1, MFN2, TOMM20, NP and actin in A549 cells infected (or not), with influenza A H1N1 virus at MOI 1 for 24h.
(TIF)

**S7 Fig. Mito-C treatment does not induce cell death. a**, A549 cells treated for 48h with increasing and indicated concentrations of Mito-C, DMSO (vehicle) or culture media without vehicle were stained with propidium iodide (PI) and analyzed by cytometry (N = 3). **b**, Lactate dehydrogenase (LDH) enzyme release was measured in supernatants of A549 cells treated for 48h with increasing concentrations of Mito-C or DMSO (N = 3). For evaluating significance of differences observed in a and b, a two-tailed Student's *t* test (NS for non-significant).
(TIF)

**S8 Fig. Growth kinetics under multi-cycle conditions. a**, A549 cells were infected at an MOI of 0.005 PFU/cell with A/Victoria/3/75 virus and treated with Mito-C (2μM), Oseltamivir (50nM) or DMSO. At the indicated times post-infection, viral titers were determined by standard plaque assay on MDCK cells. The results are shown as the mean ± SD of three independent experiments.
(TIF)

**S9 Fig. Mito-C treatment inhibits vh-RNA induced mitochondria elongation and potentiates IFN expression. a**, A549 cells, transfected with vh-RNA and treated with Mito-C at 2μM or DMSO, were immunostained with anti-TOMM20 antibody (green) and DAPI (blue) 24h post-transfection. All scale bars = 10μm. **b**, Quantification of mitochondrial elongation (TOMM20 signal) from single cells illustrated in (a) (N = 50 cells from three independent experiments) **c**, RT-qPCR analysis of IFNλ1 mRNA from A549 cells transfected (or mock transfected) with vh-RNA and treated with Mito-C at 2μM or with DMSO, at 6h or 24h post transfection (N = 3). **d**, RT-qPCR analysis of IFNβ mRNA from A549 cells transfected (or mock transfected) with vh-RNA and treated with Mito-C at 2μM or with DMSO, at 6h or 24h post transfection (N = 3For evaluating significance of differences observed in c, e, g, i, k and l, two-tailed Student's *t* test was used (*** indicates p<0.0001).
(TIF)

## Acknowledgments

We thank our team of colleagues at ENYO Pharma and at Institut Necker Enfants Malades for fruitful discussions and constant support. We acknowledge the Institut Necker Enfants Malades and Necker campus associated facilities (SFR Necker INSERM US24, CNRS UMS 3633). We also acknowledge the Centre Technologique des Microstructures (Université Claude Bernard, Lyon) and PLATIM (Plateau d'imagerie/microscopie, Lyon) and MRI (Montpellier Ressources Imagerie) platforms for imaging facilities. We acknowledge the PICT-IBiSA, member of the France-BioImaging national research infrastructure. We thank Dr. Nicolas Guyon-Gellin and Diane Sampson for critical reading of the manuscript.

## Author Contributions

**Conceptualization:** Irene Pila-Castellanos, Diana Molino, Ivan Mikaelian, Patrice Codogno, Jacky Vonderscher, Eric Meldrum, Caroline Goujon, Etienne Morel, Benoit de Chassey.

**Formal analysis:** Irene Pila-Castellanos, Diana Molino, Ivan Mikaelian, Laurène Meyniel-Schicklin, Cédric Delevoye, Caroline Goujon, Etienne Morel, Benoit de Chassey.

**Investigation:** Irene Pila-Castellanos, Diana Molino, Joe McKellar, Laetitia Lines, Juliane Da Graca, Marine Tauziet, Laurent Chanteloup, Cédric Delevoye, Olivier Moncorgé.

**Methodology:** Irene Pila-Castellanos, Diana Molino, Joe McKellar, Laetitia Lines, Juliane Da Graca, Marine Tauziet, Laurent Chanteloup, Cédric Delevoye, Olivier Moncorgé, Etienne Morel, Benoit de Chassey.

**Project administration:** Etienne Morel, Benoit de Chassey.

**Supervision:** Etienne Morel, Benoit de Chassey.

**Writing – original draft:** Irene Pila-Castellanos, Diana Molino, Laurène Meyniel-Schicklin, Eric Meldrum, Etienne Morel, Benoit de Chassey.

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
