## [Decision Letter · Decision Letter 0]

28 Aug 2020

Dear Dr de Chassey,

Thank you very much for submitting your manuscript "Mitochondrial morphodynamics alteration induced by influenza virus infection as a new antiviral strategy" for consideration at PLOS Pathogens. As with all papers reviewed by the journal, your manuscript was reviewed by members of the editorial board and by several independent reviewers. Though the topic and findings are interesting, there are several questions raised by reviewers should be addressed. In light of the reviews (below this email), we would like to invite the resubmission of a significantly-revised version that takes into account the reviewers' comments.

We cannot make any decision about publication until we have seen the revised manuscript and your response to the reviewers' comments. Your revised manuscript is also likely to be sent to reviewers for further evaluation.

Sincerely,

Shin-Ru Shih

Associate Editor

PLOS Pathogens

Ana Fernandez-Sesma

Section Editor

PLOS Pathogens

Kasturi Haldar

Editor-in-Chief

PLOS Pathogens

orcid.org/0000-0001-5065-158X

Michael Malim

Editor-in-Chief

PLOS Pathogens

orcid.org/0000-0002-7699-2064

Reviewer's Responses to Questions

**Part I - Summary**

Reviewer #1: In this manuscript, the authors report that influenza infection of cultured cells promotes an elongation of the mitochondrial network that leads to an alteration of the endoplasmic reticulum-mitochondria contact sites. Importantly, detection of influenza RNAs through RIG-I is sufficient to induce this phenotype. Finally, using a new pro-fission compound (Mito-C) that restores mitochondrial morphodynamics but interestingly reduces influenza replication, the authors demonstrate that the anti-viral effect of Mito-C is linked to the RIG-I mediated signaling at the mitochondria.

Reviewer #2: The authors of this manuscript found that infection of influenza virus promoted mitochondria elongation and the contact between ER and mitochondria. In addition, the study showed that the infection caused relocalization and reduction of DRP1, which is an important protein involved in fission of mitochondria, and increased OPA1 signal at mitochondria. Furthermore, the authors demonstrated that mitochondrial fusion could facilitate influenza viral replication, because Mito-C treatment, which could induce mitochondrial fragmentation, reduced virus production. Finally, since the antiviral activity of Mito-C was lost in RIG-I knockout cells, the authors claimed that the antiviral activity of Mito-C was dependent on RIG-I signaling pathway. Because mitochondria play an important role in antiviral signaling, the findings of this research are interesting in the field. However, several points should be further clarified to make the research more complete.

Reviewer #3: Irene Pila-Castellanos study the impact of influenza virus (Flu) infection on mitochondrial dynamics. They show that Flu triggers mitochondria elongation and alter endoplasmic reticulum-mitochondria tethering. The mitochondrial fission associated protein DRP1 is relocalized to the cytosol. Mito-C, a pro-fission molecule, restores mitochondrial morphodynamics and reduces Flu replication. Mito-C activity is linked to RIG-I signaling at mitochondria. This is a nicely presented, well controlled study.

I have only one minor comment.

**Part II – Major Issues: Key Experiments Required for Acceptance**

Reviewer #1: This paper is interesting and the conclusions raised by the authors are well supported by the data.

My only concern is about the fact that the authors report, based on IF imaging, that the influenza infection seems to induce a strong mitochondria-to-cytosol relocalization of DRP1. Unfortunately, this observation can not be confirmed by WB after cell fractionation because the viral infection surprisingly promotes a degradation of DRP1. Based on their results, it appears that the RIG-I-MAVS pathway regulate the mitochondrial dynamics. How do the authors explain that the RIG-I-MAVS pathway controls the DRP1 stability at the mitochondria? Does DRP1 undergo post-translational modification(s) in the context of influenza infection? In agreement with a previous report (PMID: 19066474), the authors observed an increase in autophagy after infection, so is it possible that DRP1 is degraded through autophagy as described recently for MAVS (PMID: 31304625)? If it is not due to autophagy, does a proteasome inhibition prevent this DRP1 degradation?

In figure 2E, the infection increased OPA1 detection and the authors propose that it is likely a consequence of a coalescence of this protein in the mitochondria. In agreement, OPA1 aggregation have been demonstrated to be required for its pro-fusion activity (PMID: 31292547). This may be mention in the text. Have the authors investigated by WB the different isoforms of OPA1 during infection?

Reviewer #2: 1. The authors found that influenza A viral RNA could be a stimulator to induce mitochondrial elongation. The result also suggested that stimulation of RIG-I signaling pathway could cause this phenotype.

(1) Does the viral RNA also reduce the contact sites of Mito/ER, like the H1N1 infection?

(2) In the experiments, the authors used 87-mer 5’ triphosphated hairpin RNA derived from an H1N1 virus. The authors should describe the sequence of the 87-mer RNA in the Materials and Methods section.

2. The results showed that infection of H1N1 virus relocalized DRP1 and decreased DRP1 expression, whereas the OPA1 signal at mitochondria was increased in the infected cells.

(1) The authors should show a blot for at least one viral protein in Fig. 2C.

(2) Is the expression of OPA1 changed in the infected cells? The authors may demonstrate it by a Western blot.

(3) In addition to OPA1, did the authors check the expression of Mitofusin 1 and 2, which are also involved in mitochondrial fusion?

3. The authors used Mito-C to induce mitochondrial fragmentation and found that the virus production was inhibited by the treatment.

(1) Does overexpression of DRP1, which might increase mitochondria fission, inhibit influenza A virus replication?

(2) Does knockdown of DRP1 expression increase replication of the virus?

(3) The infection experiment was designed for multiple-cycle replication in Fig. 3b and c. The authors may also show the virus titers at multiple time points to draw the conclusion.

4. Interferons lambda 1 and beta induced by influenza A virus infection were increased under Mito-C treatment. Can similar results be obtained in the cells stimulated with viral hairpin RNA? Mito-C treatment may be included in the experiments of Suppl. Fig. 4.

5. The authors should also determine viral titers in Fig. 4d, not only detecting the NA activity.

6. Overall, in this study, the authors tried to correlate mitochondrial morphodynamics (fission/fusion) alteration to RIG-I/interferon antiviral response and influenza virus replication. Does other signaling molecules activating RIG-I pathway, such as overexpression of RIG-I card domain, enhance mitochondrial elongation? Or is this phenomenon influenza virus-specific? The authors may also apply Sendai virus infection in the experiments of Fig. 4.

Reviewer #3: 1. Most of the experiments are performed in A549 lung cells, with one experiment showing the same phenotype in MDCK canine cells. It would be nice to show the effect of Flu infection on mitochondrial shape in other relevant human cells (cell lines or primary cells)

**Part III – Minor Issues: Editorial and Data Presentation Modifications**

Reviewer #1: In the discussion, the authors wrote that the mitochondrial elongation is detrimental to innate immunity. I guess that this point should be more discussed, for instance is it an inhibition of MAVS-mediated signaling by disruption of the ER-mito contact sites that may be important as suggested by Gale’s group (PMID: 21844353) or as a consequence of an alteration of MAVS aggregates signalosome at the surface of the mitochondrion etc?

Sup Fig 1. You wrote TRYSPIN instead of TRYPSIN.

Reviewer #2: 1. As the authors mentioned, “PB1-F2 of influenza A virus induces mitochondrial network fragmentation”. The authors may check whether the influenza A virus used in this research could produce mitochondria-targeted PB1-F2 and discuss the role of PB1-F2 in the finding.

2. In the introduction, the authors described “…. viral PB2 protein as a third viral target (Abraham et al., 2020).” Actually, Baloxavir targets on viral cap-dependent endonuclease of PA protein (Noshi et al., Antiviral Research 2018). The authors may modify the text in the section and add the “volume and pages” to the reference of Abraham et al., 2020. Clin. Infect. Dis.

3. It would be suggested to label “viral hairpin RNA” instead of “dsRNA” in the Suppl Fig. 4.

Reviewer #3: (No Response)

PLOS authors have the option to publish the peer review history of their article (what does this mean?). If published, this will include your full peer review and any attached files.

Reviewer #1: **Yes: **Damien Arnoult

Reviewer #2: No

Reviewer #3: **Yes: **Olivier Schwartz
---

## [Decision Letter · Decision Letter 1]

27 Jan 2021

Dear Dr de Chassey,

We are pleased to inform you that your manuscript 'Mitochondrial morphodynamics alteration induced by influenza virus infection as a new antiviral strategy' has been provisionally accepted for publication in PLOS Pathogens.

Best regards,

Ana Fernandez-Sesma

Section Editor

PLOS Pathogens

Ana Fernandez-Sesma

Section Editor

PLOS Pathogens

Kasturi Haldar

Editor-in-Chief

PLOS Pathogens

orcid.org/0000-0001-5065-158X

Michael Malim

Editor-in-Chief

PLOS Pathogens

orcid.org/0000-0002-7699-2064

Reviewer Comments (if any, and for reference):

Reviewer's Responses to Questions

**Part I - Summary**

Reviewer #1: In this manuscript, the authors report that influenza infection of cultured cells promotes an elongation of the mitochondrial network that leads to an alteration of the endoplasmic reticulum-mitochondria contact sites. Importantly, detection of influenza RNAs through RIG-I is sufficient to induce this phenotype. Finally, using a new pro-fission compound (Mito-C) that restores mitochondrial morphodynamics but interestingly reduces influenza replication, the authors demonstrate that the anti-viral effect of Mito-C is linked to the RIG-I mediated signaling at the mitochondria.

Reviewer #2: The authors have performed the suggested control experiments and carefully addressed the questions raised previously. The manuscript has been modified accordingly.

Reviewer #3: the authors have nicely addressed the reviewers' comments

**Part II – Major Issues: Key Experiments Required for Acceptance**

Reviewer #1: The referees' comments have been addressed in a really satisfactory manner, I therefore recommend acceptance for publication

Reviewer #2: (No Response)

Reviewer #3: NA

**Part III – Minor Issues: Editorial and Data Presentation Modifications**

Reviewer #1: (No Response)

Reviewer #2: (No Response)

Reviewer #3: NA

PLOS authors have the option to publish the peer review history of their article (what does this mean?). If published, this will include your full peer review and any attached files.

Reviewer #1: **Yes: **Damien Arnoult

Reviewer #2: No

Reviewer #3: **Yes: **Olivier Schwartz

---

## [Editor Report · Acceptance letter]

13 Feb 2021

Dear Dr de Chassey,

We are delighted to inform you that your manuscript, "Mitochondrial morphodynamics alteration induced by influenza virus infection as a new antiviral strategy," has been formally accepted for publication in PLOS Pathogens.

Best regards,

Kasturi Haldar

Editor-in-Chief

PLOS Pathogens

orcid.org/0000-0001-5065-158X

Michael Malim

Editor-in-Chief

PLOS Pathogens

orcid.org/0000-0002-7699-2064